# Making the Invisible Visible—Strategies for Visualizing Underground Infrastructures in Immersive Environments

**Sebastián Ortega** [1,*], **Jochen Wendel** [2], **José Miguel Santana** [1], **Syed Monjur Murshed** [2], **Isaac Boates** [2], **Agustín Trujillo** [1], **Alexandru Nichersu** [2] **and José Pablo Suárez** [1]

1   Centro de Tecnologías de la Imagen (CTIM), Universidad de Las Palmas de Gran Canaria,
    35017 Las Palmas de G.C., Spain; josemiguelsantananunez@gmail.com (J.M.S.);
    agustin.trujillo@ulpgc.es (A.T.); josepablo.suarez@ulpgc.es (J.P.S.)
2   European Institute for Energy Research (EIFER), D-76131 Karlsruhe, Germany; wendel@eifer.org (J.W.);
    murshed@eifer.org (S.M.M.); boates@eifer.org (I.B.); nichersu@eifer.org (A.N.)
*   Correspondence: sebastian.ortegatrujillo@gmail.com

**Abstract:** Visualization of underground infrastructure in an interactive 3D immersive environment is extremely important for efficient management of city's infrastructure. This paper describes different geometric modelling approaches to illustrate appropriate visualization of such data. A multimodal prototype has been developed by exploiting different algorithms to render these invisible underground objects as part of an urban model. This prototype has been integrated in an immersive geographic information system (GIS), named MultiVis, for handheld iOS and Android devices. As a part of the study, three distinct strategies have been tested; the first is based on the use of transparencies to convey a sense of depth, the second relies on an image-space superposition of "ditches" on top of the rendered frame and the third is a world-space deformation of the elevation model that exposes the underground elements. Furthermore, a comparative user experience analysis of different techniques aimed to the geometrically accurate visualisation of utility networks and other underground facilities are performed and evaluated. It includes a set of user evaluations for different parameters of these techniques, which gives us an insight on how the proposed methods affect the experience and usability for technical and non-technical users.

**Keywords:** augmented reality; virtual reality; visualization; underground infrastructure; user experience survey

## 1. Introduction

The efficient management and sustainable development of the infrastructures in a city is a crucial task for local governments. Such infrastructures are the foundations of modern civilization, as they instantaneously support the fulfilling basic human needs even with increased pressure on them due to massive worldwide urbanization processes. Precise information and knowledge about these infrastructures is required for efficient urban planning, urban management and infrastructure resilience [1–4]. City infrastructure is thereby generally referred to structures and services that act as a basis for the economy and the quality of life in a city that can include both, hard infrastructure such as utility networks or transportation structures and soft infrastructure such as IT services. In this paper, with the term city infrastructure, we refer to utility networks such as fresh or waste water, oil, gas, electricity, communication networks and other underground installations such as manholes or distribution stations. These infrastructures are still traditionally managed using paper-based documents, written records and maps. With the emergence of information and communication

technologies, digitization and visualization techniques are used in some cities [5,6]. However, despite the rapid digitization process and the mapping of underground infrastructures many subsurface assets are still not accurately mapped or records about the existence of such structures are completely missing due to innacurate asset management (Wahab et al. [7]).

Appropriate visualization is therefore of crucial use for utility networks planning, management and maintenance, essentially supporting the entire life cycle of the infrastructure components. These infrastructures can be graphically represented using different features, e.g., line features to represent network segments, point features to illustrate connections, substations or valves with attributes and geographic locations (x and y coordinates) as well as their associated depth or height (z coordinates). Multiple standards for representing, exchanging, analysing and storing utility network data exist such as the INSPIRE Utility Networks model ([8]) that is based on the generic INSPIRE Network Model ([9]) in the european context, the ESRI Geometric Network Model ([10]) for the representation of network types within the ArcGIS environment or Industry Foundation Classes (IFC) standard for representing utility data in the framework of Building Information Modeling (BIM) ([11]). These standards are very different in terms of detailed contents but they share similar fundamentals characteristics as they are developed with local context in mind. They describe similar components such as pipes, cables, flows which are included in all types of utility networks. A comprehensive summary of numerous utility network data standards is provided by Hijazi et al. [12]. Visualizing these different types of features is a challenging process. Most recent research on spatial visualizations has focused on the above surface context, and traditional Geographic Information Systems (GIS) were typically developed as 2D desktop GIS applications, which ignored the representation of underground features.

In the last two decades, a shift from traditional 2D desktop GIS to 3D and 4D desktop and mobile GIS applications has emerged where 3D/4D visualization is used because it allows for a more effective representation of utility network features (Zlatanova et al. [13], Du and Zlatanova [14]). Different approaches ranging from more traditional vertical profile visualizations to immersive visualization such as Augmented Reality (AR) and Virtual Reality (VR) have been developed. However, these applications did not find widespread usage until recently; mainly due to the lack of accessible and limited hardware options as well as suitable Software Development Kits (SDK) (Amin and Govilkar [15]). In 2017, the development and introduction of more performant consumer mobile hardware and native software support from both Google, with its ARCore SDK (https://developers.google.com/ar/), and Apple, with its own ARKit SDK (https://developer.apple.com/arkit/), allowed the creation of performant applications and optimal support of cameras and sensors already built-in in the mobile hardware (Christen et al. [16]).

Furthermore, the immersive visualization of geo-data gained traction in recent years. Popular gaming engines such as Unity3D (https://unity3d.com/) and Unreal (https://www.unrealengine.com) are now supporting the integration of spatial data and are allowing the development of spatial AR and VR application via specific plugins for ARKit and ARCore. In addition, traditional GIS vendors such as ESRI recently started to include AR and VR capabilities within their GIS environments. For example, ESRI launched AR and VR capabilities within their ESRI ArcGIS Runtime SDK (http://proceedings.esri.com/library/userconf/devsummit18/papers/dev-int-212.pdf) in 2018 allowing the immersive visualization of geodata as 360 VR scenes, AR, VR and mixed reality. Multiple applications mainly focused on infrastructure data have been developed already [17,18].

The focus of research and the development of such applications is still mainly focused on hardware and software implementation aspects and less on the visualization aspects [19–22]. The contribution of this research lies in describing optimal visualization methods of underground utility data in immersive environments, which are not provided in previous literature. We introduce the MultiVis application (Supplementary Materials) and applied different geometric modelling approaches to illustrate the various underground features in immersive environment.

In the following section, an overview of current visualization research on immersive applications for utility network data is summarized. Section 3 and Section 4 describe the data infrastructure and the MultiVis application (Supplementary Materials) that has been extended and ported from iOS to Android for this research, respectively. Section 5 then describes the implementation and visualization strategies of displaying underground infrastructure while Section 6 focus on the experiment setup and the user survey. Finally, a discussion and future direction of this research follows in Section 7.

## 2. Related Work

During the last two decades, several studies have offered solutions to the accurate visualization of underground elements in Virtual or Augmented Reality applications. In this regard, a first AR prototype is introduced by Roberts et al. [23], who relies on GPS sensors and visual tracking to overlay in a Head-Mounted Display (HMD) device an underground utility network over the real scenario. Bane and Hollerer [24] proposed a "Tunnel Tool" visualisation in which data related to hidden and occluded parts of the scene are rendered inside a frustum, generating a tunnel effect in the final image. Avery et al. [25] applied an "edge overlay" effect, wherein the outlines of visually distinct features on occluding surfaces are preserved to provide depth cues and achieve an "X-Ray vision" effect.

It was around the year 2009 that two different approaches surged. The first visual representation deals with the estimation of positioning error of an underground network model. Such a technique is desirable in visualization for field technicians since the risk of damaging hidden subsurface elements is minimized during maintenance tasks. On this topic, Su et al. [26] investigated the use of an uncertainty region around the pipe geometry, which is shown in the scene as a semi-opaque polygon. Li et al. [27] empirically derived a proper size for such an uncertainty region by comparing utility network plans with in situ real-time kinematic global positioning systems (RTK-GPS) and ground-penetrating radar (GPR) sensor measurements. Zhang et al. [28] compared the precision and perceived depth appearance obtained after placing the underground network in an AR app with computer vision matching techniques and sensor-based techniques. Finally, olde Scholtenhuis et al. [29] showed a fuzzy 3D-model of the utilities in an AR app for smart glasses and tablets. Models containing cylindrical halos were used to represent minimum, mean and maximum placement error margin.

The second strategy seeks to improve the way in which underground objects are represented in an AR application. In this regard, Schall et al. [30] proposed an excavation tool, which simulates a hole on the ground in which the pipeline is visible. Their work was later extended [3] so that the application could assist with the maintenance tasks of underground utility networks. They also compared their excavation tool with trench-like and shadow-like representations. Chen et al. [31] combined the "X-Ray vision" approach with aperture focus and context concepts to extract the depth order and mobile elements of a scenario to generate blending masks to draw the occluded objects. Finally, Zollmann et al. [32] compared the use of alpha-blending, edge "ghosting" and image-based ghosting techniques for representing subsurface objects in images.

Most underground-related applications are thought to be only useful in the context of AR and when designed for geological purposes (Lee et al. [33]) or management of power and water underground utilities [3,27,29,34]. However, for multimodal applications we find the example of [35], in which a unique application features multiple environments, including virtual globes and VR or AR viewers. Their research shows that integrating underground visualization is still an open problem. This work aims to propose a visualization scheme of subsurface/underground objects that remains useful for all of the aforementioned integrated view modes. In this regard, we also illustrate different use cases using appropriate underground data and AR/VR technologies.

### 3. Data Infrastructure for Representing Subsurface Features

In holistic contemporary urban planning processes, such as in the context of Smart Cities applications, underground infrastructure is usually not analysed and displayed exclusively but it is commonly interpreted as a part of a larger urban model including buildings, pump stations or reservoirs (Delmastro et al. [36], Li et al. [37]). While different data standards for utility data such as the INSPIRE Generic Network Model ([9]), the ISO standard Industry Foundation Classes (IFC) ([11]) or the ESRI Geometric Network model ([38]) have been developed before, most of them are lacking in explicit control over all of the necessary fundamental aspects required to fully model physical, functional and semantic properties of arbitrary utility networks in a three-dimensional context (Kutzner and Kolbe [39]). An early decision to use one of these data models as the basis for a subsurface utility network feature visualisation application may therefore lead to consequent conceptual and technical limitations. This risk becomes exacerbated as such an application grows and is applied to networks of increasing physical, functional or semantic diversity and/or complexity.

CityGML is an international Open Geospatial Consortium (OGC) standard for the representation and exchange of semantic 3D city and landscape models. It allows for standardized abstraction and exchange of urban objects and their relationships in the form of semantic city models. Its data model is based on the ISO 19100 standards family and it is implemented as an application schema for OGC's Geography Markup Language (GML) [40]. The Application Domain Extension (ADE) allows for extension of the CityGML standard for the purpose of modelling urban objects pertaining to a specific theme. The Utility Network ADE is one such extension that offers new urban objects and properties pertaining to utility networks. While still in development, it has been shown to be mature enough to model the constituent features of a real urban multi-network system, as well as functional and connective relationships within and between networks [41–44].

With specific regards to visualisation, the precise, explicit and abstract nature of the utility network data model for CityGML schema uses hierarchical inheritance-based relationships to make common properties shared between utility network features, while each feature defines its own unique properties. For example, a *RoundPipe* element and a *RectangularPipe* element are both children of the *AbstractPipe* element, and can therefore share properties implemented by the *AbstractPipe* element, such as material type, intended function, construction year, etc. However, the *RoundPipe* feature implements properties pertaining to diameter, whereas the *RectangularPipe* feature implements properties pertaining to width and height. In this research a *RoundPipe* model is used as they are the most vastly used on civil infrastructures (Figure 1). These inheritance-based relationships allow for individual features to be stored in a manner which lends itself well to a standardisation of visualisation processes.

In this paper, the Utility Network ADE was used to model a sample multi-network comprised of different kinds of pipes in a technological park called Technologiepark, situated in Karlsruhe, Germany. Using this data model resulted in a simple data sample for rendering the properties of different kinds features in different of networks, while ensuring that subsequent developments of rendering ability can be tested simply by expanding the data sample. Expanding our data sample is, for the reasons discussed above, straight forward and any use case of a visualization application can likely be addressed. Furthermore, the direct link to the core CityGML data model supports future integration with smart city applications.

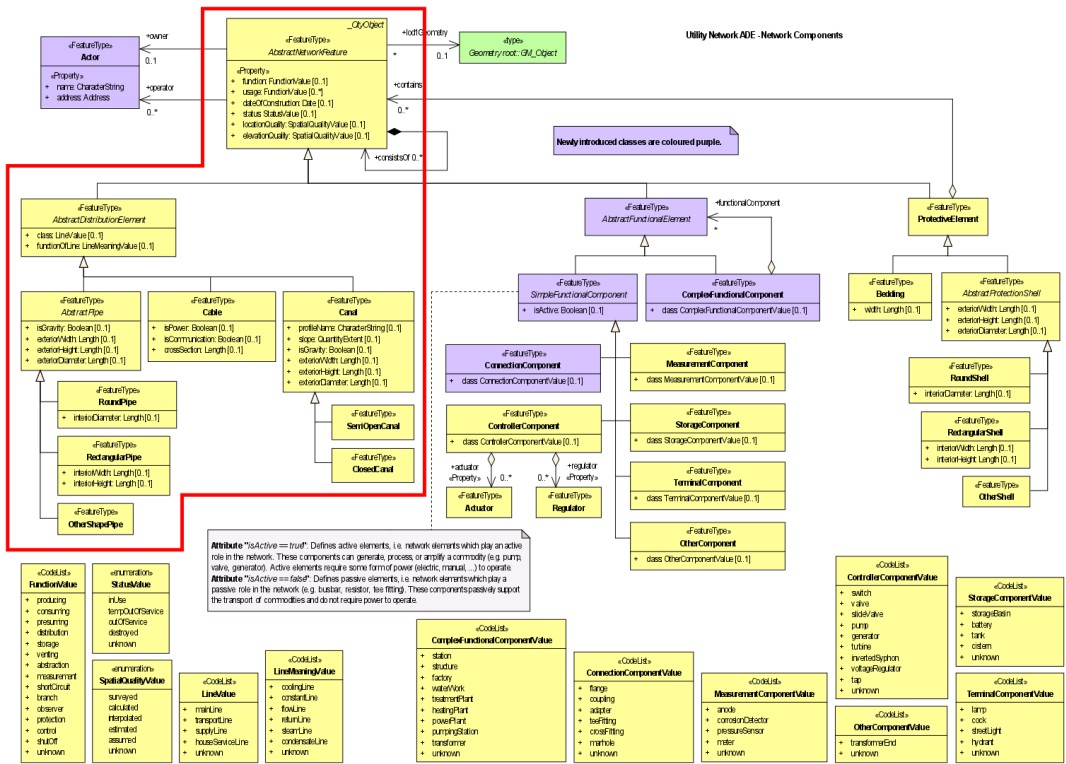

**Figure 1.** UML diagram of the Utility Network ADE Network components [45]. The highlighted area depicts the elements used for rendering and visualization.

## 4. Implementation of the MultiVis Mobile Application

The MultiVis application (Supplementary Materials), previously presented in Santana et al. [35] and in Wendel et al. [46] is used for the implementation and testing of different visualization modes for depicting utility network data in this research. The application focuses on a holistic approach that implements a seamless transition between a traditional virtual map view to virtual and augmented reality (VR and AR) modes in a single mobile application. This is of particular interest to experts and decision makers as it provides them with means to explore results on-site through AR or VR. These two visualization techniques have been combined with traditional maps for a better overview and strategic planning capabilities (Wendel et al. [46]).

MultiVis has been implemented using the Glob3 Mobile (G3M) framework (Santana et al. [35]). This SDK, presented in Suarez et al. [47], is a mobile-oriented framework for the development of map and 3D globe applications, being highly configurable in terms of user navigation and level of detail (LoD) strategies. Thus, the framework is suitable for the present research, having recently demonstrated several possibilities that mobile devices offer for the planning of complex infrastructures and working with large datasets.

The G3M API allows the generation of map applications in 2D, 2.5D and 3D following a zero third-party dependencies approach and provides native performance on its three target platforms (iOS, Android, HTML5). 3D graphics are supported by the Khronos Group application programming interfaces (APIs), OpenGL ES 2.0 on portable devices and WebGL (web counterpart of OpenGL) on the HTML5 version. The features of this framework include multi-LoD 3D rendering and automatic shading of objects. Due to the multiplatform nature of the G3M, API portability to Android or HTML5 is possible. Figure 2 shows the different visualization modes that were adopted from Santana et al. [35] and ported to Android.

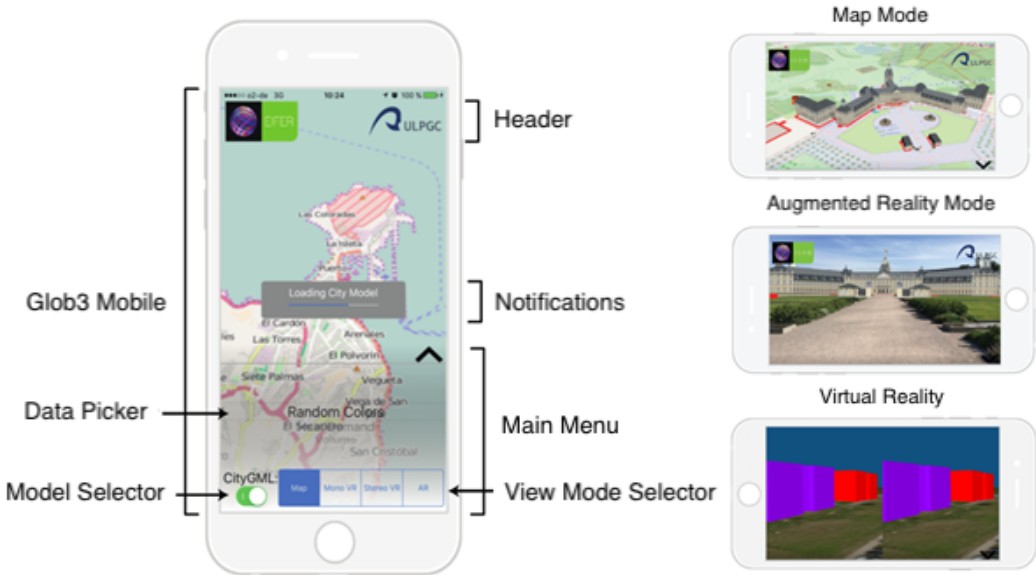

**Figure 2.** User interface layout for iOS and the three implemented visualization modes.

## 5. Geometrical Modelling and Rendering

This section describes the modelling of the network tessellation and its subsequent rendering process. The result of such a tessellation is a mesh which represents all the round pipes which belong to the utility model and another one which contains surrounding ditches described in Section 5.2.2. Both meshes are stacked on the final image by means of a multi-pass rendering system, as discussed in Section 5.3. Figure 3 describes different states and sub-stages of this complete process.

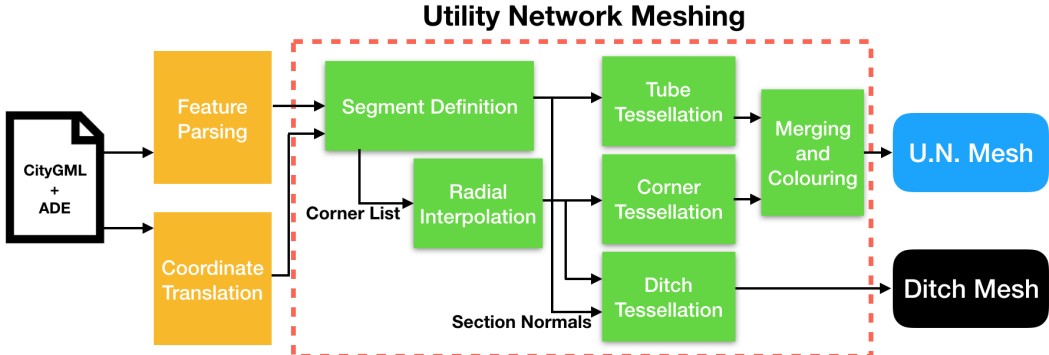

**Figure 3.** Flow chart of the utility network mesh generation process. RoundPipes are extracted from the CityGML model (orange) for a subsequent meshing (green). The process generates a list of straight segments and rounded corners that are tessellated independently. The final product is a set of tubular shapes signalizing the pipes (Utility Network Mesh) and surrounding black terrain trenches (Ditch Mesh). See Figure 10 for wireframe representation.

### 5.1. Generation of Pipes from Line-String Based Networks

Commonly, underground round pipes are characterized as a linked chain of straight segments and a radius, *r*. This geometry is presented in our model accompanied by metadata regarding the inner and outer materials of the pipes. As precondition to the tessellation process, the model must be previously cleaned of consecutive segments in the same direction, which are merged beforehand. Each one of these round pipes is visually represented by a tubular mesh of its outer surface.

The first step consists in refining the corners of the pipe polyline, which normally do not represent a smooth bending at the corners. To that end, each segment in the set is shortened at both ends by

the same length, *r*, which is the radius of the cylinder. A cross-section of the tube is generated at each segment end by rotating a point at a distance r in the direction of the normal of the segment. The point is then rotated *n* times $(360/n)°$, with *n* being the desired number of vertices per segment, as it could be seen in Figure 4, left.

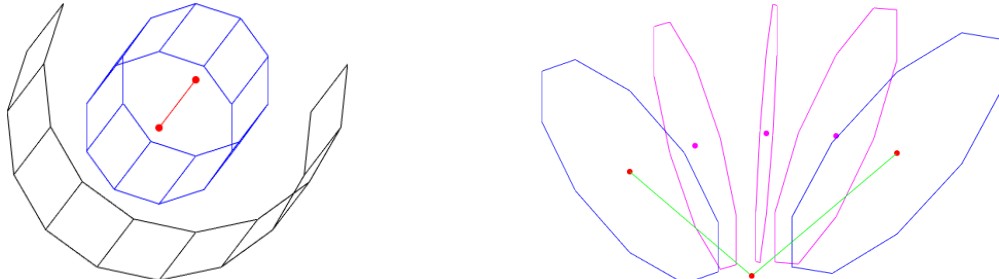

**Figure 4.** Generation of cylinders from a segment (**left,** blue) and rounded corners between two cylinders (**right**, magenta).

During the smoothing of each bend, the rotation needed to bring each end of a segment to the start of the next is computed. The pivot point of such rotation is computed as the intersection of both segment normals. Once the rotation between segment ends is computed, *m* new intermediate covers are generated by rotating $\gamma/(m-1)$ (being $\gamma$ the angle of the joint) the end cross-section of the first segment. This is best shown in Figure 4, right. Finally, the tubular surface is subsequently generated by connecting the vertices of all the computed cross-sections via triangles. The final mesh is stored as a triangle strip to improve the rendering performance.

In this work, a utility network data set was created that depicts common features present in the study area in Karlsruhe, Germany. Figure 5 shows the utility network data consisting of utility pipes of different diameters, depths and types.

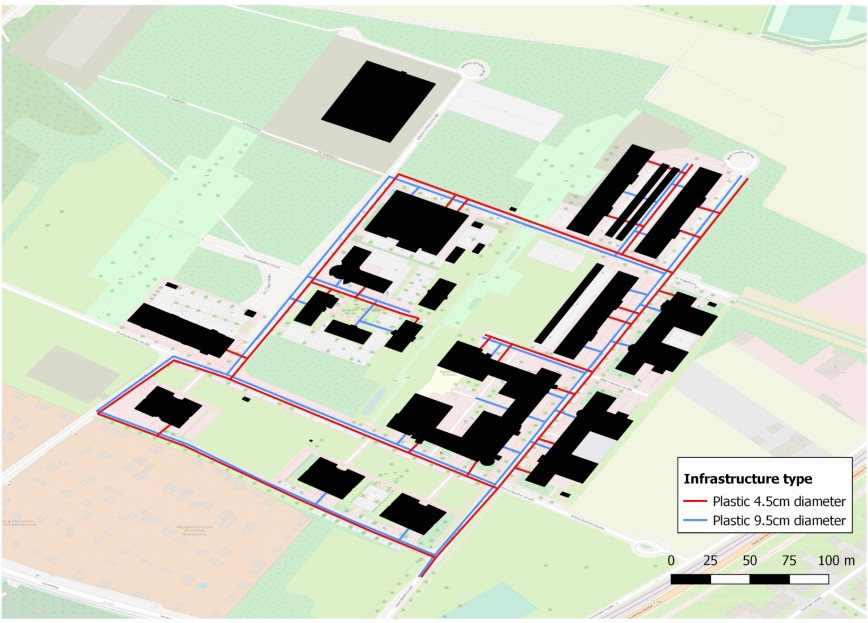

**Figure 5.** Spatial extent of the study area with data curated from OpenStreetMap and a utility network in the CityGML Utility Network ADE standard.

## 5.2. Strategies for Visualization of Underground Utilities

The meshed model generated in the previous section has been rendered during our research following different strategies, so they can be compared in terms of user experience. Four strategies, which are summarized in Figure 6, have been explored in this work: variable static, alpha-blending, ditches and an excavation tool.

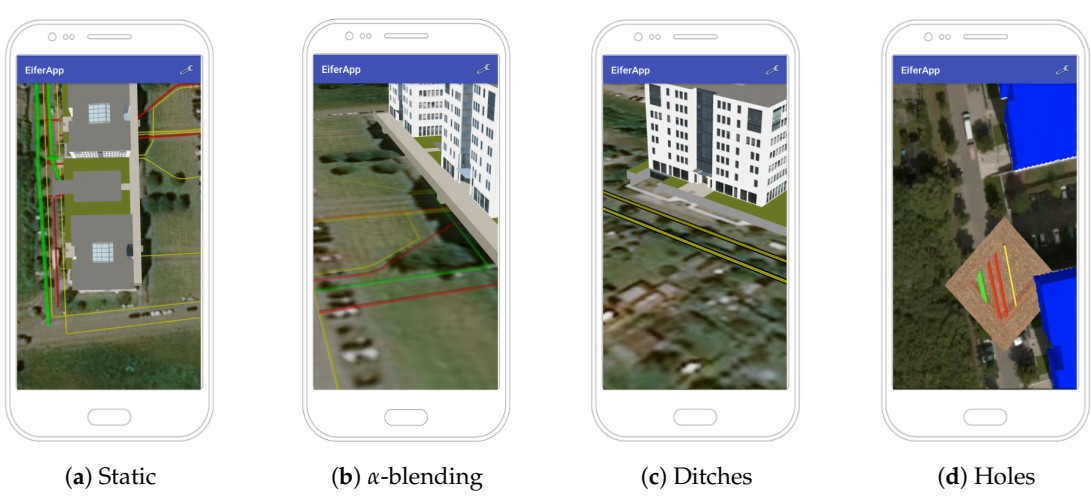

| (**a**) Static | (**b**) $\alpha$-blending | (**c**) Ditches | (**d**) Holes |

**Figure 6.** Strategies for visualization of underground utilities.

### 5.2.1. Variable $\alpha$-Blending

The use of transparencies to indicate that an object is below the surface is widely extended upon from previous works. A simple manner of this effect is to apply a single value of transparency ($\alpha$) to the whole object, regardless of how deep or how far from the user viewpoint it is. Zollmann et al. [32] considered the option of applying different transparency values for underground elements, but did not implement the technique, focusing on ghostings instead. Moreover, which function should be used to assign each in order to convey the sensation of depth is another issue to be considered.

In this regard, our proposal is to make the function dependent on the distance to the viewer, $D_i$, applied to each pipe vertex closer than a maximum allowed distance $D_{max}$, as depicted in Figure 7. Considering $D_{max}$, $D_i$ can be remapped into the range $[0-1]$. Let the result of the remapping be $d_i$. This allows for $\alpha$ values to be assigned according to the following expression:

$$\alpha_i = 0.5 \cdot (1 - f(d_i)) \tag{1}$$

in which $f(d_i)$ can be any function which returns a value in the range $[0-1]$ for any given $d_i$. Since the choice of $f(d_i)$ will affect the final visualization, several alternatives have been explored. Table 1 summarizes the different alternatives and their corresponding equations. Figure 8 shows the variation of these equation for ranges of $d_i$, between 0 and 1.

The transparency of each rendered fragment of the model is computed at a per-frame basis. However, in order to improve efficiency, the results are stored in graphics memory, so calculations are only done on changes of the virtual camera position. Once all values are calculated, the color information per vertex, including $\alpha$, is sent to a shader engine to apply the final appearance of each feature.

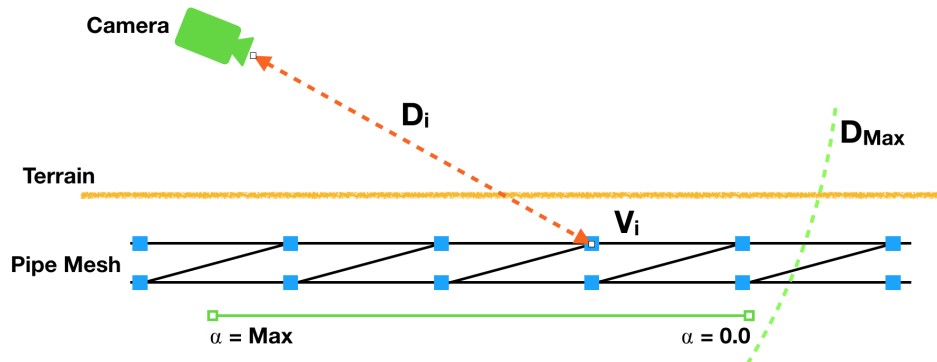

**Figure 7.** Pipe transparency is a function from the distance to the camera to the pipe mesh, computed at the vertices ($V_i$). $D_{Max}$ is the maximum distance at which the transparency value ($\alpha$) should be 0.

**Table 1.** Different distance-based $\alpha$ subfunctions.

| $f(d_i)$ | Expression |
|---|---|
| Fixed | $0$ |
| Linear | $d_i$ |
| Smoothstep [48] | $3d_i^2 - 2d_i^3$ |
| Logistic [49] | $1/(1 + e^{-s_i})$, where $s_i = 10d_i - 5$ |
| tanh | $0.5 + 0.5 \cdot tanh(s_i)$, where $s_i = 10d_i - 5$ |
| arctan | $\frac{1}{3} \cdot (arctan(s_i + 1.5))$, where $s_i = 20d_i - 10$ |
| Softsign [50] | $0.5 \cdot \left(\frac{s_i}{1+|s_i|} + 1\right)$, where $s_i = 100d_i - 50$ |

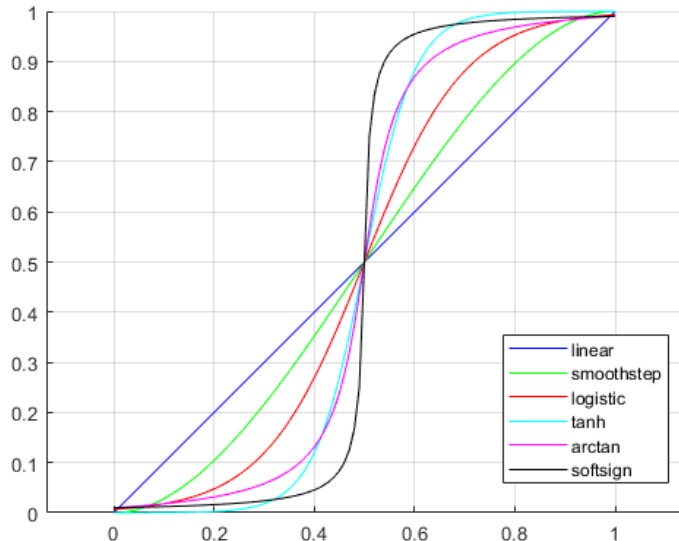

**Figure 8.** Superposition of the different tested expressions, with values of $d_i$ mapped into the range $[0, 1]$.

### 5.2.2. Visualization of Underground Pipes via Terrain Ditches

As an alternative to using transparencies, the creation of a second geometry model to add an underground context is proposed. This model, called *ditch*, is semi-cylindrical and surrounds the lower half of the pipe. The ditch is rendered in the scenario with a flat and dark color, so the pipe seems to be placed inside and below the ground plane, as seen in Figure 9.

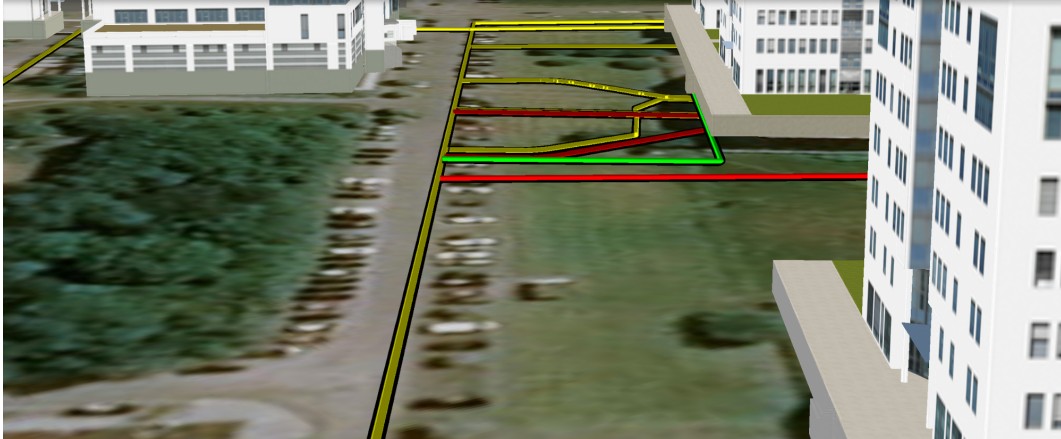

**Figure 9.** Pipes and their ditches in a 3D scenario simulating a street.

The generation of the ditch model is analogous to that of the pipe, creating semi-cylinders and rounded corners based on the line segments, as well as a radius. This allows the creation of both meshes following a parallel process, as it is appreciated in Figure 10, saving computational resources.

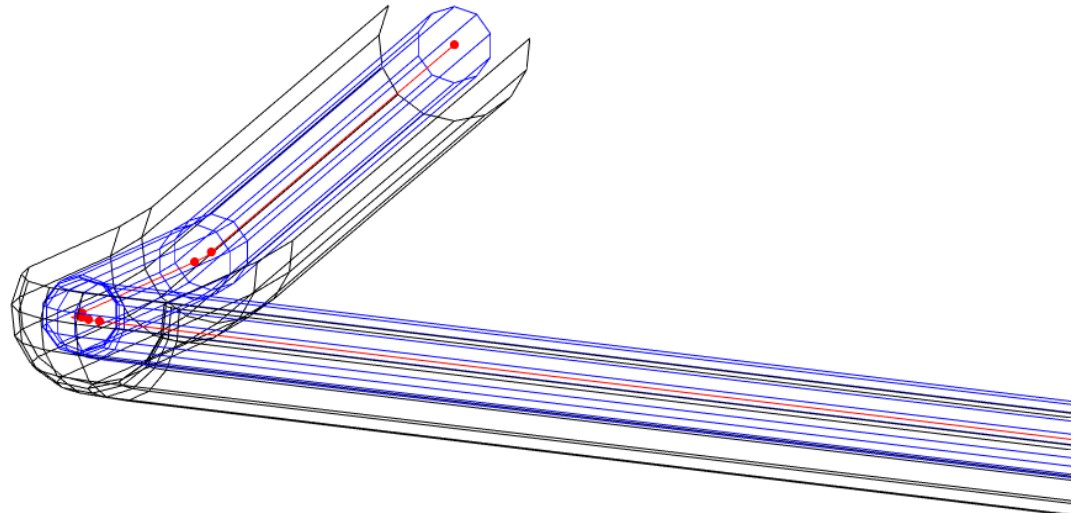

**Figure 10.** Generation of a ditch model (black) surrounding the pipe (blue) generated for an underground network object (red).

### 5.2.3. Excavation Tool

The last rendering strategy considered in this work is to extend the proposal of an excavator tool, which was done for AR by Schall et al. [30]. It could be used not only in AR-based applications, but also in virtual globe-based ones. The placement of the hole is dependent on user interface events.

For maps, the hole generation is triggered by a user command. In our implementation, the user long-presses a screen pixel, which is then counter-projected to the scenario. If the outgoing ray intersects a position of the terrain, it is used as pivot for the hole generation. In a VR environment, the hole is generated at a fixed distance in front of the user. The hole is then stationary, while the user can walk, take a look and inspect the revealed underground elements.

To take full advantage of the tridimensional nature of this kind of scene, a rectangular area of the *Digital Elevation Model* (DEM) is modified in our implementation, so its height is $D$ meters deeper than the shallower point in that area. To preserve the continuity of the terrain, skirts [51] are generated whenever necessary. Finally, a textured mesh is created in the edges of the hole region to simulate the walls of the hole. This can be seen in Figure 11.

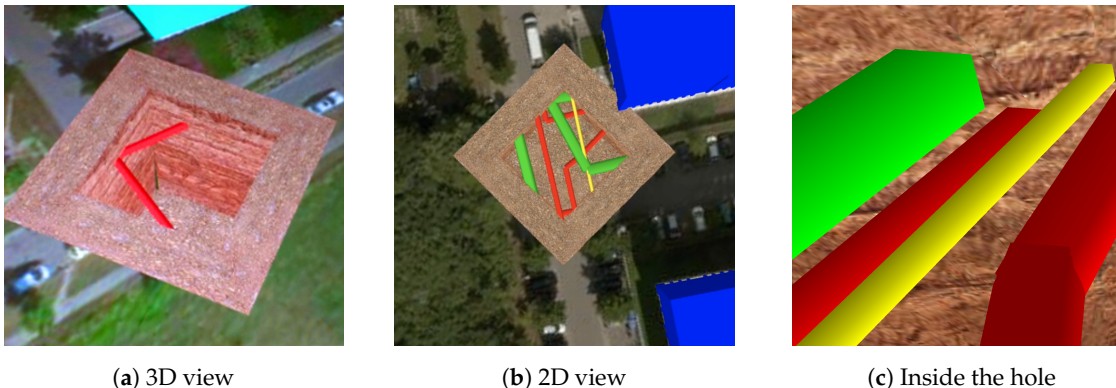

| (**a**) 3D view | (**b**) 2D view | (**c**) Inside the hole |

**Figure 11.** Underground network visualization using the excavator tool extension [3] for virtual globes.

Although depth cues are improved for map and virtual reality by using this technique, some issues must be addressed before using it in a multimodal application. For instance, a *level of detail* (LoD) becomes necessary. On one hand, for large scale rendering, the LoD may be coarse enough for the hole visualization to be imperceptible. A possible and straightforward strategy is to make the region affected by the hole larger as the level of detail becomes coarser. At the same time, a LoD strategy should also be implemented to generate new hole elevation models for these cases. Another possibility is to enable user interaction on the "excavation", allowing it to move and change its size according to gestures in the screen. These options will be studied in future extensions of this work.

### 5.3. Rendering of Underground Elements

Rendering a scene with many detailed and diverse geo-referenced elements, like those typically seen in an urban environment, requires sending enormous amounts of information, such as vertices, textures, indices, colors and illumination conditions, all of which contribute to the final rendered image. Moreover, if the scene contains a great number of these assets, careful management of the graphics information is absolutely necessary in order to avoid redundancies.

G3M allows the addition of elements like markers, meshes or shapes to a virtual map, and it is designed to automate the selection of a proper shader program for any given symbol and manage its associated transactions and data storage [52].

To do so, all the necessary rendering data are introduced in a *Directed Acyclic Graph* (DAG), like the one in Figure 12. Using this structure, the rendering settings and pipeline information presented as uniforms and attributes, which are common to multiple renderable objects, are contained in the parent nodes of the DAG. This way, data redundancy is reduced. During the creation of a given frame, the DAG is traversed in depth, and all available information across the path to a leaf node is gathered. After that the resulting data set is processed, generating all the necessary inputs for the graphics pipeline. An example on how it will work for three geometry symbols representing a pipe, a ditch and a hole wall is introduced in Figure 13.

According to these inputs, a program should be selected from a shader library integrated in the framework. The system selects the shader that best matches the available rendering data for each DAG path. Finally, a value checking prevents re-sending redundant information during the graphics data transfer. Depending on the type and repetition of the symbology, this technique could save up to 80% of the transfers [53] and speedup the process, reusing the data and the program lookup in consecutive frames.

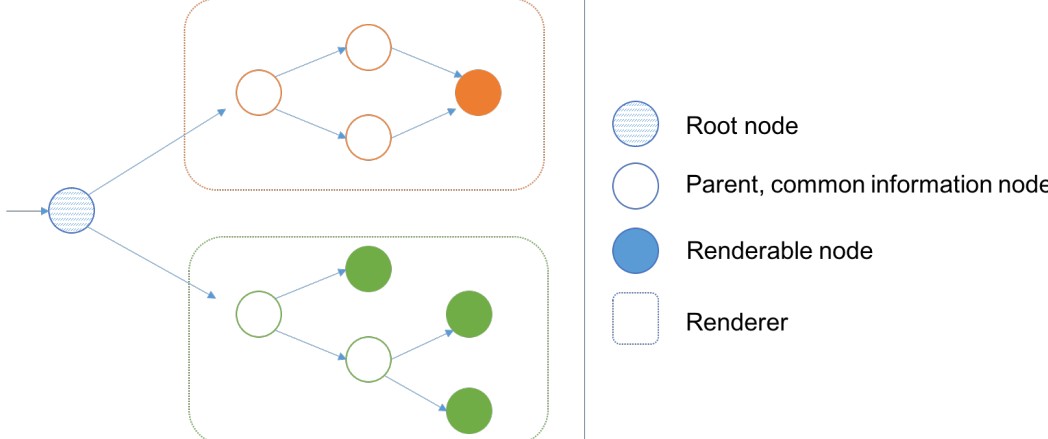

**Figure 12.** A directed acyclic graph to efficiently generate inputs for graphics pipeline from symbology. Nodes could be accessed via multiple paths. Parent nodes contain pipeline information common to many renderable nodes. Different colors mean different *Renderers* within the G3M framework.

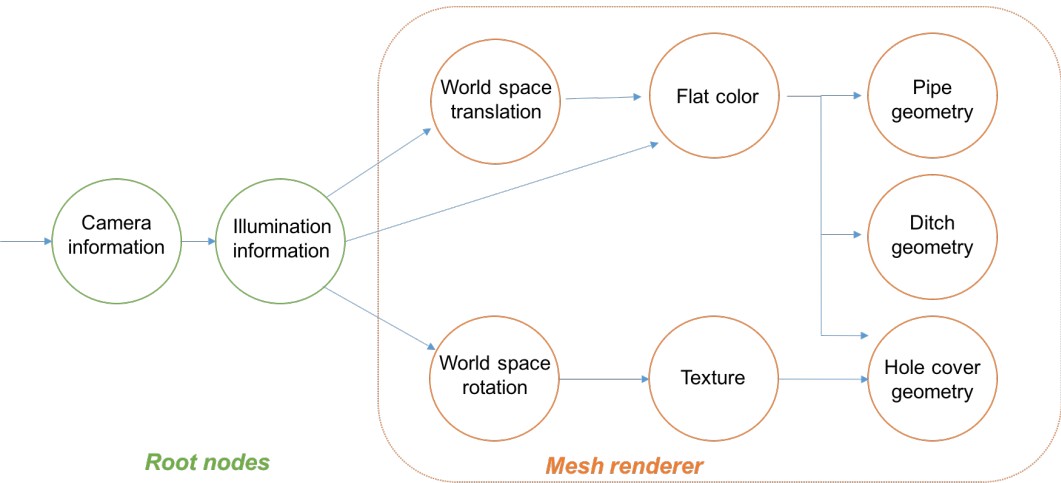

**Figure 13.** Direct acyclic graph for an example where 3 geometry symbols (pipe, ditch and cover) require 7 symbol instances.

Each one of the different initial branches in the DAG directly corresponds to a certain manager component, a *Renderer* [54], which takes care of a specific kind of symbology. As an example, the *PlanetRenderer* is the branch with information about terrain and map imagery, the *MarkRenderer* processes 2D markers, the *MeshRenderer* deals with vertices, indexes, textures and other things needed to generate meshes like the underground pipes, etc. During the rendering, *PlanetRenderer*, the module in charge of terrain model generation and rendering, is always processed first. The content included in the rest of the renderers are then processed in a user-defined order.

A limitation of this scheme is exposed when underground elements are included in the scene. If a depth test is performed, underground elements will not be shown in screen. If depth test is disabled for those particular elements, they will be rendered on top of every above-ground object, losing all depth cues and potentially confusing the users. This is the starting point for the alpha-blending strategy (Section 5.2.1). For ditches (Section 5.2.2), a multi-pass rendering scheme [55], also called layered rendering, was also implemented on the G3M engine.

To achieve the desired effect of terrain transparency, the abovementioned multi-pass rendering strategy renders different layers of the scenario with independent depth buffers. Renderers containing the underground symbology are set as second-pass renderers. In case there are over-surface elements that could eventually occlude the underground network (e.g., building models), their renderers should also be marked as second-pass renderers.

At the time of rendering, the planet and first-pass content are drawn first. After that the *Depth Buffer* is cleared. This will allow us to use the depth test with underground information. Finally, the elements contained in second-pass renderers are drawn over the same *Frame Buffer* in the pre-established order.

## 6. Experimentation and User Survey

To compare the behaviour of the different proposed strategies and determine the one which results in the most convenient visualization, a survey has been conducted with participants applying a methodology similar to the one introduced in the work of Mirauda et al. [34]. The participants were required to navigate through the MultiVis application (Supplementary Materials) in a 3D scenario, which includes representations of objects over the surface (buildings, trees, sensors, etc) and objects below the surface. To ensure that all participants went through all visualization options, they were personally instructed and went through a written guideline on how to use the app. The participants tested the application and its settings using the Karlsruhe TPK study area (Figure 5). An overview of the survey questions is provided in the appendix of this paper (Appendix A).

The survey participants are asked whether the pipe network looks like it is placed below the surface or not, for all the functions described in Section 5.2.1, as well as the ditch and excavation strategies described in Sections 5.2.2 and 5.2.3. Responses are encoded so that their answers fit into a 5-point Likert scale. A value of 1 indicates that the participant felt like the feature does not look like it is underground at all. A value of 5 indicates that the participant felt like the feature was certainly underground. Additionally, users were invited to experiment with different $D_{max}$ values for the function of their choice and indicate the value which, in their opinion, made the visualization more useful. Extra controls have been included in the Multivis interface for the survey, and they are shown in Figure 14:

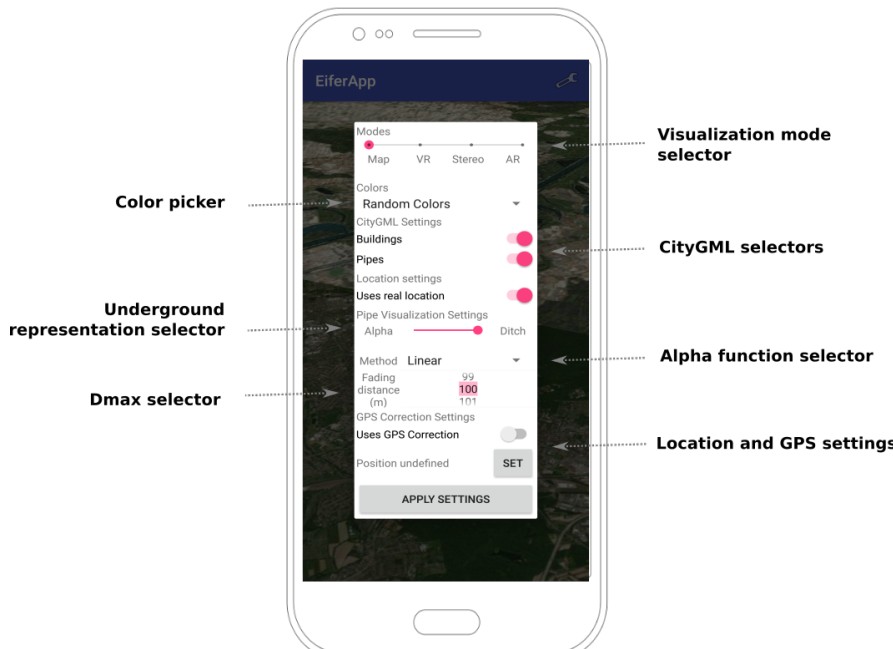

**Figure 14.** User interface layout for the survey in the Android platform. Underground-related selectors are added to the control panel.

30 participants were interviewed for the study. All genders and ages between 16 and 75 are represented in the sample. 17 participants declared they had previous technical experience. The answers of this questionnaire, organized by method, are introduced in Figure 15.

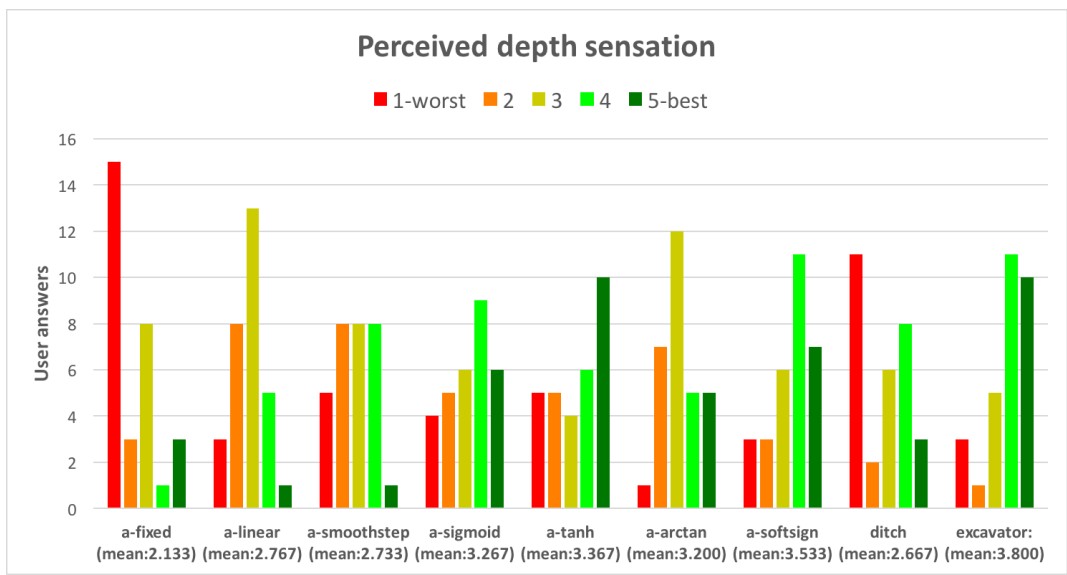

**Figure 15.** Answers of the survey participants per method.

The results of the survey reflect that the use of new techniques for underground visualization improves the depth cues obtained from using a fixed $\alpha$ blending. From the explored methods, the best-received one by the survey participants was the excavation tool, with an average rating of 3.8 and a median of 4. From the variable $\alpha$ functions, survey participants tended to prefer the softsign function (average rating 3.53, median 4). Other options, like sigmoid and tanh, were also appreciated, but there was more variability of perceived effectiveness among the test subjects. The ditch technique was, on average according to the survey participants, the least effective method, again with an average rating of 2.67, a standard deviation of 1.47 and a high variability in the responses. With respect to the ditch methodology, a trend was also observed in the responses.

As Table 2 shows, participants without technical experience had more difficulty in understanding the ditch technique and tended to interpret what they are seeing as over-surface structures, while most of the technical users found it effective. Considering that tools which deal with underground utility information are normally intended for people who have technical experience and work in the field, the ditch technique could still be a possibility to present the underground information in a multimodal application for which the target user group is one with significant technical experience. The excavation tool and the variable alpha blending appear to be the most generally accessible methods. However, these results suggest that a more thorough investigation should be conducted on how the previous technical information of users affects their comprehension of a 3D scene showing 3D subsurface network elements.

**Table 2.** Variability on the validation of ditch technique between users with and without previous technical experience.

|  | People | Mean | Median | Standard Deviation |
|---|---|---|---|---|
| Technical users | 17 | 3.235 | 4 | 1.393 |
| Non technical users | 13 | 1.923 | 1 | 1.256 |

With respect to the most appropriate $D_{max}$, the mean of the responses was 844.44 m, with a standard deviation of 972.84 m. However, almost two thirds of the participants prefered a distance between 100 and 500 m, as it could be seen in Figure 16, which makes the median value, 300 m, an acceptable initial setting for $D_{max}$. The mode value, 200 m., is also a good initial setting for $D_{max}$ in the MultiVis app, due to its disproportionate representation in the overall responses. Although several participants indicated they preferred to maintain a general view of the utilities in the area,

and therefore chose a $D_{max}$ of more than a kilometer. This fact suggests that a multimodal application which wants to take advantage of a variable $\alpha$ blending method for visualization should offer this option as a user-adjustable parameter.

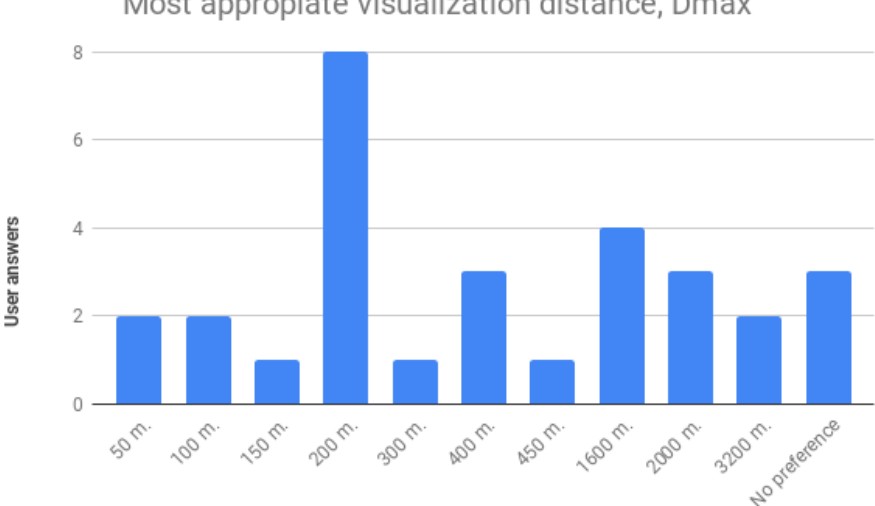

**Figure 16.** Distribution of preferred $D_{max}$ distances.

## 7. Conclusions and Further Work

In this research, several alternatives for the visualization of underground structures have been discussed for its use in a multimodal application. An alpha blending strategy has been discussed, implementing a set of functions that map the viewer-object distance to the transparency of each feature in different ways. Furthermore, a novel technique for representation of subsurface utilities, named "ditches", has been proposed. Finally, an extension for multimodal applications of the excavation tool of Schall et al. [30] was further developed.

The behaviour of these techniques were validated by conducting a user experience survey with 30 participants in which all the visualization techniques are demonstrated to represent different underground water pipe networks in the case study located in the city of Karlsruhe (Germany). The results reflect that all techniques are more effective than the fixed transparency blending reference method. In general, users found more effective depth cues while using the excavator tool extension. The distance function which achieves better visualization results for variable transparency blending was found to be the softsign function method. This method was also rated the overall second best rated technique. Most participants preferred a maximum distance of subsurface utility network feature visualization between 100 and 500 m. Finally, a pattern in the ratings of the ditch technique suggested that the previous knowledge and experience of the users may affect their comprehension of the visualization of subsurface utility networks.

The results gathered in this research need also to be analyzed and evaluated from a use case dependent perspective. The original objective in the first development of the MultiVis app had originally been seen as a proof of concept for the integration of CityGML spatial data and the G3M library for 3D data visualisation to test weather the CityGML data standard is suitable for multimodal visualizations. As the application, presented in Santana et al. [35], has now matured from a prototype to a stable app we envisioned creating distinct use cases to technical and non-technical users. In the case of the ditch method for representation we received positive feedback from the technical users which have experience working in the field, showing the clear need for this kind of representation of subsurface infrastructure for the expert user.

At the time of writing, the prototype and techniques introduced still present some limitations for the rendering of some elements present on underground networks. Namely, the connection of underground pipes and cables with buildings and other above ground structures has to be factored into

the rendering pipeline. Finer detail models would also be of use for photorealistic applications, which would have to be produced by the tessellation system. At this point, due to the lack of higher detailed city models, only Level of Detail (LoD2) models were used that represent themselves too coarse for realistic representations especially in the VR mode where the user has to be fully immersive into the scene. In the same way, for use cases that require high positional accuracy levels, GPS positioning may lack the required accuracy. We are currently working with visual odometry tools and beaconing devices to overcome this issue and to provide higher positional accuracy of the app.

Future extensions of this work could include a deeper analysis on how the technical experience of the user could affect the understanding of the scene, improvements for the visualization of ditches and holes, as seen in Figures 17 and 18 respectively, and the management of user interactions (see Figure 19) to better adapt the excavation tool to the needs of multimodal applications. Additional user experience surveys will be necessary for determination of enhancements of the user experience and usability of the interaction with the app.

Besides these further technical implementations, it will be necessary to establish and extend current cartographic rules for the representation of underground features. From our literature review and from the review of current research and commercial applications we found that most of the applications that are showing underground infrastructure are lacking on the visualization side.

Current cartographic guidelines and visual variables are lacking the support for underground infrastructure visualization. Existing cartographic principles should be adopted to the usage and visualization of underground infrastructure in outdoor environments in immersive ways. While Becker and König [56] already presented first representation strategies for utility network data they are still lacking visualization guidelines for utility networks in immersive environments. The establishment of such new rules will be beneficial for further developers to develop better and more intuitive visualizations and applications.

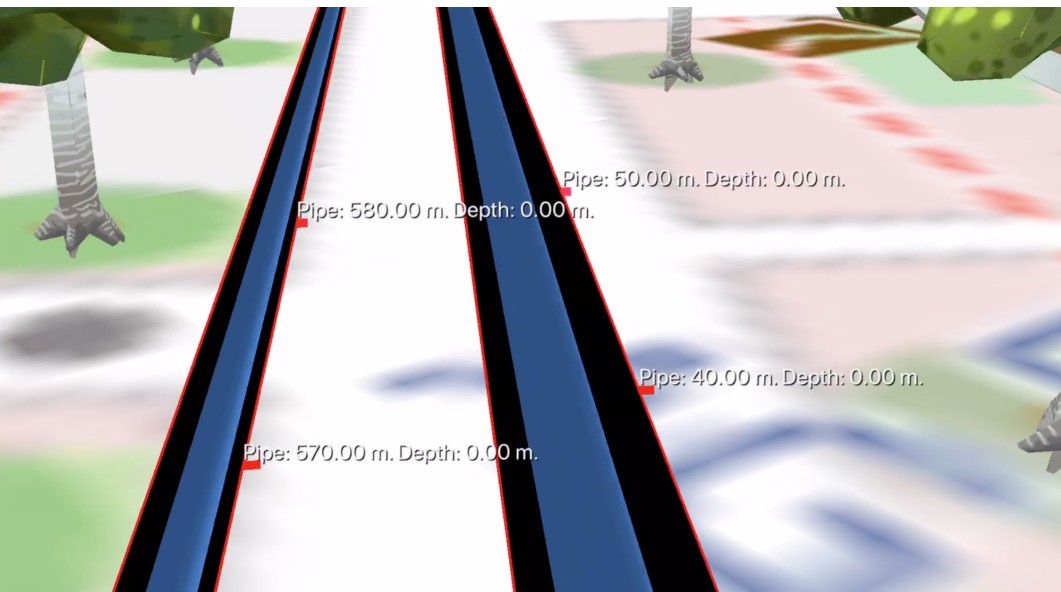

**Figure 17.** Ditch highlighted with borders to enhance depth perception. Evenly spaced tick marks along pipes and cables also provide further information to the user.

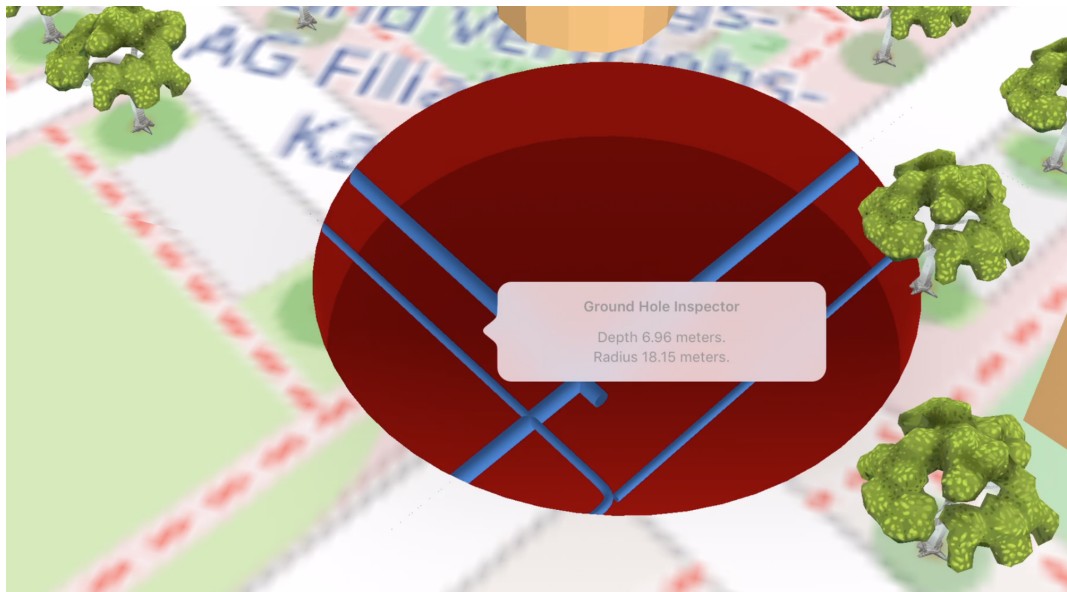

**Figure 18.** Different ground hole styles are being tested as future development of the project. In the picture a "cartoonish" rounded hole for pipe inspection.

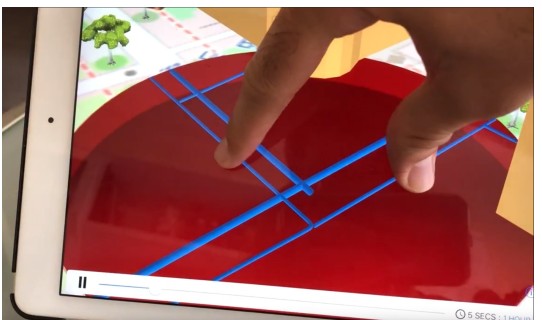 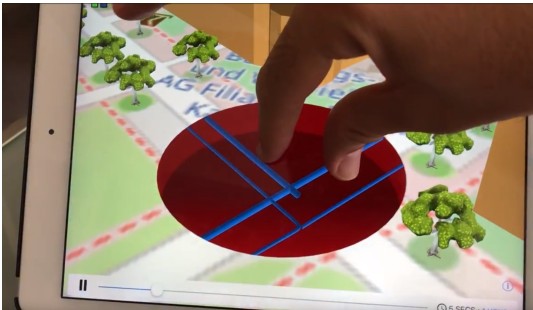

**Figure 19.** Multitouch composition of ground hole, to enable the interactive inspection of underground objects.

**Supplementary Materials:** A supplementary demo video of the MultiVis application is available on https: //www.youtube.com/watch?v=7XczNhw-lwg.

**Author Contributions:** Conceptualization: Sebastián Ortega, Jochen Wendel, José Miguel Santana, Agustín Trujillo; Methodology: Jochen Wendel; Software Development: Sebastián Ortega, José Miguel Santana; Validation: Jochen Wendel, Syed Monjur Murshed; Formal Analysis: Sebastián Ortega; Investigation: Sebastián Ortega, José Miguel Santana, Jochen Wendel, Isaac Boates; Resources: Syed Monjur Murshed, Agustín Trujillo, José Pablo Suárez; Data Curation: Isaac Boates, Agustín Trujillo, Writing—Original Draft Preparation, Sebastián Ortega, Jochen Wendel, José Miguel Santana, Isaac Boates; Writing—Review and Editing: all authors; Project Administration: Syed Monjur Murshed, Jochen Wendel, Agustín Trujillo, José Pablo Suárez; Funding Acquisition: Syed Monjur Murshed, Jochen Wendel.

**Funding:** The 1st author would like to thank the Universidad de Las Palmas de Gran Canaria for its grant "Programa de personal investigador predoctoral en formación 2015", which made possible his work. We are also grateful to EIFER/EDF R&D for partial funding of this research.

**Acknowledgments:** The authors would like to thank the participants of the user survey in Las Palmas de G.C., Spain and Karlsruhe, Germany, members of the Escuela Universitaria en Informática (ULPGC) and the European Institute for Energy Research (EIFER) respectively. Furthermore, the authors would like to thank Alexander Simons (EIFER) for database and PostGIS CityGML support, Alberto Pasanisi (EIFER), Andreas Koch (EIFER) and the N41 EIFER group for support in the user surveys and for reviewing the paper and improve the overall quality of this research.

**Conflicts of Interest:** The authors declare no conflict of interest.

**Appendix A. Survey Questionnaire**

You should open the app and navigate to the TPK - Karlsruhe area, where some pipes are placed. You can move through the area by using a finger, do zoom and unzoom by using two fingers and change between a 2D and a 3D view by using three fingers. Before answering any question, always move through the utility of your choice and change the view from 2D and 3D. 1-to-5 options always go from 1, the worst possible sensation of underground, to 5, the best possible sensation of underground.

1. Initially, utilities are visualized in the app with the Ditch technique. Please use the fingers for navigating to a pipe of your choice. Does it look as it is placed underground? (Options: 1-2-3-4-5)
2. Use the menu to change the visualization mode to Alpha with method Fixed. Please go to another pipe of your choice. Does it look as it is placed underground? (Options: 1-2-3-4-5)
3. Please set now the method to Linear in the menu and navigate to another pipe of your choice. Does it look as if it is placed underground? (Options: 1-2-3-4-5)
4. Please set now the method to Smoothstep in the menu and navigate to another pipe of your choice. Does it look as if it is placed underground? (Options: 1-2-3-4-5)
5. Please set now the method to Sigmoid in the menu and navigate to another pipe of your choice. Does it look as if it is placed underground? (Options: 1-2-3-4-5)
6. Please set now the method to Tanh in the menu and navigate to another pipe of your choice. Does it look as if it is placed underground? (Options: 1-2-3-4-5)
7. Please set now the method to Arccos in the menu and navigate to another pipe of your choice. Does it look as if it is placed underground? (Options: 1-2-3-4-5)
8. Please set now the method to Sigmoid in the menu and navigate to another pipe of your choice. Does it look as if it is placed underground? (Options: 1-2-3-4-5)
9. Change now the visualization mode to Hole, and do a long-press over an area to excavate it and see the utilities on it. Do they look as if they are placed underground? (Options: 1-2-3-4-5)
10. Finally, we are looking for the best maximum distance for showing utilities in the scenario. Please set the alpha visualization mode and the linear method. Now, please test different fading distances and then use the fingers to zoom in, zoom out, move and change to 3D view. We suggest testing 50, 100, 200, 400, 800, 1600 and 3200, but you could also test a distance of your choice. Which one is, in your opinion, the most appropriate for the visualization of utilities? (Options: A number)

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
