# Peer review of "Making the Invisible Visible—Strategies for Visualizing Underground Infrastructures in Immersive Environments"

_ijgi, doi:10.3390/ijgi8030152_

Round 1

Reviewer 1 Report

The presented research constitutes a valuable development of the current state of knowledge on the visualization technology of the utilities network. The authors have properly reviewed important scientific studies in this topic. Establishing research in the context of past achievements develops the state of knowledge. The article is well built, although as an original scientific paper should contain a separate chapter on materials and methods and organization of research, which are scattered throughout the text, even in the introduction. In addition, it is not understandable in the text: "name some standards here please; or 37 just mention that Section 3 gives detailed description "(36 p.2). This is a fair working remark that has not been taken into account by the authors, but it is worth to refer to this.
I recommend the print article after considering the comments.

Author Response

Dear Reviewer 1

thank you for your comments, we have addressed all of your comments.

We have corrected the grammar and spelling issues addressed in the review.

We have also added more clarification to the comment raised on line 36, we have added the following:

"Multiple standards for representing, exchanging, analyzing and storing utility network data exist such as the INSPIRE Utility Networks model (JRC 2013a) that based on the generic INSPIRE Network Model (JRC 2013b) in the european context, the ESRI Geometric Network Model (ESRI 2019) for the representation of network types within the ArcGIS environment or Industry Foundation Classes (IFC) standard for representing utility data in the framework of Building Information Modeling (BIM) ((ISO 16739:2013)"

Regarding the comment of adding a new section on methods and materials.

We have added more content to the introduction section in order to be more precise in the introduction on the methods.

Reviewer 2 Report

The manuscript considers visualization of subsurface utilities. The authors present software (app) that can be developed to aid visualization for different practical applications.

I am not qualified to judge on software soundness and accuracy.

Regarding a research problem itself, subsurface visualization, insufficient background is given. I recommend to expand Introduction mentioning sate-of-the-art in underground space planning and mapping, and GPR use as well. Authors should take in the account that information on many subsurface assets can't be extracted from databases.

Recommended references\reading:

Bobylev N, Hunt DVL, Jefferson I, Rogers CDF, (2013) Sustainable Infrastructure for Resilient Urban Environments. In: Advances in Underground Space Development – Zhou, Cai & Sterling (eds), Copyright 2013 by The Society for Rock Mechanics & Engineering Geology (Singapore). Published by Research Publishing. pp. 906 – 917. ISBN: 978-981-07-3757-3; doi:10.3850/978-981-07-3757-3 RP-107-P219

Wahab, S.W., Chapman, D.N., Rogers, C.D.F., Foo, K.Y., Metje, N., Nawawi, S.W., Isa, M.N., Madun, A. (2018) Assessing the condition of buried pipe using ground penetrating radar (GPR). International Archives of the Photogrammetry, Remote Sensing and Spatial Information Sciences - ISPRS Archives, 42 (4/W9), pp. 77-81. DOI: 10.5194/isprs-archives-XLII-4-W9-77-2018

Hojjati, A., Jefferson, I., Metje, N., Rogers, C.D.F. (2017) Embedding sustainability criteria into pre-appraisal of underground utility for future cities. Proceedings of the Institution of Civil Engineers: Urban Design and Planning, 170 (6), pp. 258-271.

Overall the research question is topical and interesting to wider audience and I recommend publishing after revision.

thank you.

Author Response

Dear reviewer 2

thank you for your comments and valuable input.

We have incoperated all your comments addressed in the review

Thank you for your recommendation on the additional literature. We think this is a valuable additons and we have added the following sentences to the Introduction section to include the new literature.

"Precise information and knowledge about these infrastructures is required for efficient urban planning,  and urban management and infrastructure resilience (Hojjati et al. 2017, Bobylev et al. 2013, Schall et al. 2013, Becker et al. 2012)"

"However, despite the rapid digitization process and the mapping of underground infrastructures many subsurface assets are still not accurately mapped or records about the existence of such structures are completely missing due to inaccurate asset management (Wahab et al. 2015)."

Reviewer 3 Report

The authors describe three methods of presenting visualizations of underground infrastructure developed on top of MultiVis GIS renderer for handheld devices.

The authors describe the underlying platform as well as the development of the visualization methods reasonably well. There are minor grammar mistakes but presentation is mostly clear.

The main drawback of the manuscript is the description of the user tests. The current description is rather vague and does not describe what the users did exactly and how they answered the questions. Were the users allowed to roam freely and try the visualizations at their own pace? If not, what was the protocol? How was it ensured that users went through all visualization options?

The authors should do a spell check and fix any remaining grammar mistakes and unclear sentences. For example, line 36 looks like it has a comment that looks like it was not meant for the final manuscript.

Author Response

Dear Reviewer 3,

thank you for your valuable input. We have addressed the comments made in the user testing section.

We did the following changes to the document:

We added the following sentences to be more precise on how the user testing was conducted

The participants were required to navigate through the MultiVis application in a 3D scenario, which includes representations of objects over the surface (buildings, trees, sensors, etc) and objects below the surface. In order to ensure that all participants went through all visualization options, they were personally instructed and went through a written guideline on how to use the app. The participants tested the application and its settings using the Karlsruhe TPK study area (Figure 5). An overview of the survey questions is provided in the appendix of this paper (A1).

For more clarification we added the user survey questions in the Appendix section of the paper

We have also address the grammar and spelling issues in the document